# National administrative record linkage between specialist community drug and alcohol treatment data (the National Drug Treatment Monitoring System (NDTMS)) and inpatient hospitalisation data (Hospital Episode Statistics (HES)) in England: design, method and evaluation

Emmert Roberts [1,2,3,4] James C Doidge,[5] Katie L Harron [6]
Matthew Hotopf,[2,3] Jonathan Knight,[4] Martin White,[4] Brian Eastwood,[4]
Colin Drummond[1,3]

BE and CD contributed equally.

For numbered affiliations see end of article.

**Correspondence to**
Dr Emmert Roberts;
emmert.roberts@kcl.ac.uk

## ABSTRACT

**Objectives** The creation and evaluation of a national record linkage between substance misuse treatment, and inpatient hospitalisation data in England.

**Design** A deterministic record linkage using personal identifiers to link the National Drug Treatment Monitoring System (NDTMS) curated by Public Health England (PHE), and Hospital Episode Statistics (HES) Admitted Patient Care curated by National Health Service (NHS) Digital.

**Setting and participants** Adults accessing substance misuse treatment in England between 1 April 2018 and 31 March 2019 (n=268 251) were linked to inpatient hospitalisation records available since 1 April 1997.

**Outcome measures** Using a gold-standard subset, linked using NHS number, we report the overall linkage sensitivity and precision. Predictors for linkage error were identified, and inverse probability weighting was used to interrogate any potential impact on the analysis of length of hospital stay.

**Results** 79.7% (n=213 814) people were linked to at least one HES record, with an estimated overall sensitivity of between 82.5% and 83.3%, and a precision of between 90.3% and 96.4%. Individuals were more likely to link if they were women, white and aged between 46 and 60. Linked individuals were more likely to have an average length of hospital stay ≥5 days if they were men, older, had no fixed residential address or had problematic opioid use. These associations did not change substantially after probability weighting, suggesting they were not affected by bias from linkage error.

**Conclusions** Linkage between substance misuse treatment and hospitalisation records offers a powerful new tool to evaluate the impact of treatment on substance related harm in England. While linkage error can produce misleading results, linkage bias appears to have little

### Strengths and limitations of this study

► This record linkage represents the first study of its kind to link centralised national level substance misuse treatment data and inpatient hospitalisation records.
► No single unique identifier, such as National Health Service number, is routinely collected within the National Drug Treatment Monitoring System and fewer personal identifiers are routinely collected than in other UK government held datasets.
► The limited availability of personal identifiers results in an increased risk of both false and missed matches, which could potentially affect the validity of any subsequently conducted analyses.
► Linkage error did not appear to lead to systematic bias and misestimation of sociodemographic and clinical factor associations with the average length of hospital stay.

effect on the association between substance misuse treatment and length of hospital admission. As subsequent analyses are conducted, potential biases associated with the linkage process should be considered in the interpretation of any findings.

## INTRODUCTION

Routinely collected administrative data from the health and social care sector is increasingly used to both inform public health policy, and to generate research. While several initiatives across the UK have used national record linkages to further population-level

understanding in specific disease areas,[1] the ambition to harness record linkage as a means of improving health outcomes for people with drug and alcohol misuse has not been fully realised.

The number of people accessing specialist alcohol treatment has fallen by 19% between 2013 and 2017, while the number hospital admissions in which alcohol was recorded as a contributory factor has increased by 5% in the same timeframe.[2–4] Given this context, a recent report from the UK Department of Health and Social Care identified an urgent need to estimate the impact of specialist drug and alcohol treatment on acute care resource usage and substance-related harm.[5] The report posits that this goal may be achieved through detailed analysis of linked individual-level hospitalisation and substance misuse treatment data, which could '… generate evidence to quantify the impact on health services utilisation before and after successful treatment'.[5]

The National Drug Treatment Monitoring System (NDTMS) is the centralised database, collated and maintained by Public Health England (PHE), which receives monthly input from all local authority commissioned community drug and alcohol services in England.[6] This contains individual-level data on an individual's sociodemographic characteristics (date of birth (DOB), gender, ethnicity, housing status etc), diagnostic characteristics, including the quantity and frequency of individuals' substance use, and treatment characteristics including frequency and type of contact with treatment services, the interventions received, and measures of treatment success. Hospital Episode Statistics (HES) is the centralised repository, collated and maintained by National Health Service (NHS) Digital, which collects all information pertaining to NHS hospitalisation in England and Wales.[7] The HES Admitted Patient Care (APC) database is one of the main administrative databases operating under the umbrella of HES and covers all NHS inpatient admissions, including any admission to private or third sector hospitals subsequently reimbursed by the NHS.[8] As such HES APC is estimated to contain >99% of all inpatient hospital activity in England.[9] An inpatient hospital admission includes any secondary care-based activity requiring a hospital bed, thus includes day cases, and both planned and emergency admissions, in physical and mental health settings. HES APC does not cover accident and emergency (A&E, emergency department) attendances, nor outpatient bookings, these data being held in separate HES databases.

Although NDTMS has been previously linked with mortality data from the Office of National Statistics, and the Police National Computer,[10–12] the lack of linkage between NDTMS and inpatient hospitalisation data limits the capacity to evaluate the impact of specialist drug and alcohol treatment on individual and regional rates of hospitalisation. International efforts have been made to facilitate record linkage of national databases in order to evaluate substance misuse outcomes,[13] however, previous studies have often lacked access to national level data on substance misuse treatment, due in part to fragmented

healthcare delivery systems, or lack of a centralised data repository. As centralised national databases exist for both hospitalisation and substance misuse treatment in England, we sought to link these two databases to inform drug and alcohol policy and research.

In this report we describe the process of record linkage and aim to evaluate the linkage quality and its potential impact on any subsequently conducted analyses. We believe this record linkage may result in the largest cross-sectional and longitudinal substance misuse database globally, and as such could become a resource which is able to support a large number of analytic outputs with the aim of improving the lives of those with substance use disorders.

## METHODS

Patients accessing publicly funded specialist drug and alcohol treatment services in England provide written consent to share their information with NDTMS, and are informed that NDTMS records may be linked with data from specifically sanctioned UK government-held databases, including HES.[14] Over 98% of patients provide consent,[15] and the nature of this consent states that any record linkage would be undertaken by PHE, and that individuals may opt out at any time from having their records used within NDTMS.

### Patient and public involvement

The study benefited throughout from discussion with the South London and the Maudsley Biomedical Research Centre Data Linkage Service User and Carer Advisory Group, and the PHE Alcohol Treatment Expert Group which includes experts with lived experience. The former group represents a regular meeting of people whom have an interest in projects involving data linkage, and who have lived experience of mental health diagnoses, including substance use disorders. They receive on-going training on data matching processes, and hence can make recommendations on the acceptability of suggested data flows. The current proposal was presented in June 2018, and there was group-wide acknowledgement of the importance of the proposed linkage, based on personal experience of treatment experiences in drug and alcohol services. The group were content with the linkage methodology proposed, including the use of patient identifiers. Both groups will remain involved in subsequent analysis plans from any resultant linked data.

### Linkage methods

Record linkage is the process of bringing together information pertaining to the same individual (or entity) from different databases. Linkage applies a set of criteria to determine whether or not records belong to the same individual, and aims to assess the true match status of each record pair: either a 'match', that is, records belong to the same individual, or a 'non-match', that is, records belong to different individuals. If record pairs

are misclassified, error may be introduced as either 'false matches', that is, records from different individuals link erroneously or 'missed matches', that is, records from the same individual fail to link. Introduction of bias from linkage error, particularly if risk factors for important outcomes are associated with error rates, can impact the validity of findings derived from linked data.[16 17] This is more likely to occur if datasets do not have a unique identifier in common.[18]

We selected all NDTMS records for adults accessing specialist drug or alcohol treatment in England between 1 April 2018 and 31 March 2019 as the test linkage population. The structure of the test linkage NDTMS data is such that one record represents one unique adult (n=268 251).[19] This was matched against all HES APC records available since database inception on 1 April 1997. The structure of the HES APC data is such that the same unique individual has multiple records, one for each hospital admission episode, with unique individuals identified by a specific variable, the HESID, which is assigned by NHS Digital (n=390 642 220 records; n=67 378 943 unique individuals).[20] As not all individuals presenting to drug and alcohol services will have been admitted to hospital, we did not expect all NDTMS records to match with HES APC. No unique person identifiers, such as NHS number, were shared between both databases but a number of personal demographic and geographic identifiers were available for matching. Identifiers were harmonised to maintain a consistent format across the two databases, which included harmonising string length, use of spaces, capitalisation and hyphens. Five variables were available for matching; an individual's DOB, sex, postcode, ethnicity and General Practitioner (GP) practice. Full variable descriptions can be found in online supplemental material.

A Structured Query Language (SQL) algorithm was designed to facilitate NDTMS to HES APC linkage. Initial data cleaning in both datasets included the conversion of all missing or non-valid data to null values and the collapse of all postcodes relating to a no fixed abode (NFA) status into a single value. All NDTMS records contained a validly coded value for DOB and sex while 96.3% had a validly coded postcode, 94.7% a validly coded ethnicity and 18.4% a validly coded GP practice. NDTMS records that had missing or invalid postcodes (n=10 011, 3.7%) were excluded from linkage, as a combination of sex, postcode and DOB was the minimum—but not necessarily sufficient—data required to uniquely identify an individual. Of the remaining n=258 240 NDTMS records n=6878 (2.7%) shared the same combination of DOB, sex and postcode, of which n=164 (2.4%) did not have a valid entry for either ethnicity or GP practice.

Matching was based on an exact match for each of the five variables described earlier, and conducted hierarchically in four stages as below:

Stage 1: exact match on DOB, sex, postcode, ethnicity and GP practice.

Stage 2: exact match on DOB, sex, postcode and GP practice.

Stage 3: exact match on DOB, sex, postcode and ethnicity.

Stage 4: exact match on DOB, sex and postcode.

When records matched, they were removed from the dataset and not included in subsequent matching stages. As both databases are longitudinal, it was possible that several different values for postcode, and GP practice were recorded for each individual over time. Where more than one unique value was available the hierarchical algorithm attempted to link NDTMS records to HES APC records sequentially starting with the most recent value for each variable. All resulting records that linked with multiple records from the other dataset were removed and treated as non-links.

### Gold-standard subsample

A small subset of people in the full NDTMS sample (n=1328), who to date were taking part in the PHE individual placement and support trial,[21] had consented to make their unique 10-digit NHS number available. As NHS number is also coded within HES APC, this was used as a single unique identifier, common to both datasets, to facilitate linkage within this 'gold-standard' sample of individuals in NDTMS who had their NHS number available. The sociodemographic and clinical characteristics of the full NDTMS sample and the 'gold-standard' NDTMS sample are available in online supplemental table S1.

Using the 'gold-standard' sample the linkage rate was calculated as the percentage of NDTMS individuals linked to any HES APC record first by exact matching on only NHS number, and second using the four-stage deterministic algorithm described previously. The results were evaluated to determine the missed match rate, and the overall linkage precision, that is, the proportion of links that are true.

Individuals linked using their NHS number were deemed to have been definitely hospitalised within their lifetime. Within this sample, individuals that were linked and not linked using the four-stage algorithm were compared with estimate rates of missed links. To allow for variation in patient characteristics and data quality between data providers, as well as between individuals, we used multilevel logistic regression with nesting of individuals within local authority commissioned treatment services and match status in NDTMS as the binary outcome (match=1, non-match=0). Model fit was examined using a likelihood ratio test comparing the multilevel model to a fixed-effects logistic model which did not account for nesting of individuals. We explored any association between match status and NDTMS sociodemographic (eg, sex, age, ethnicity, NFA status and Index of Multiple Deprivation), and clinical factors (eg, the misused substance/s for which the person entered treatment). For modelling purposes ethnicity was recoded into the binary categories of white and non-white,[22] and

probability estimates of matching as a function of the independent variables were generated.

## Analysis of linkage error

Challenges exist to assess the impact of linkage error when the outcome in question may not have been experienced by all people in the sample to be matched. When linking an individual's NDTMS records to HES APC it is difficult to know which matches have been missed as the HES database by design will only capture information about individuals who have been hospitalised. As such non-links could be due to an individual never having been admitted to hospital or being a missed match.[23 24]

For each unique linked individual, a binary outcome of their average length of hospital stay (≥5 days=1, <5 days=0) was created to assess bias due to linkage error. This was chosen as it is clinically relevant, reflecting the current UK average length of hospital stay per person, and is recorded for all people within HES APC.[7] Using the estimated probability of matching from the 'gold-standard' analysis, we created a weight that was inversely proportional to the probability of being linked to HES APC data using the four-stage algorithm. These weights were subsequently assigned to each linked individual, as per standard methods to account for non-response bias in cross-sectional and cohort studies.[25 26] Univariable multilevel logistic regression was used within the 'gold-standard' sample to examine the association between independent variables and the average length of hospital stay. Estimates were generated using the 'unbiased' linked sample matched using NHS number, and these were then compared with estimates obtained using the 'biassed' linked sample matched using the four-stage algorithm. The model applied to the 'biassed' sample was first conducted without any weighting, second conducted incorporating the inverse probability weights to examine if this corrected any linkage error and third conducted weighted according to the odds of having sufficient matching data.

## Data access

While access to the linked dataset is only available within PHE, subject to approval, extracts of NDTMS are available to researchers through the Office of Data Release at PHE,[27] and extracts of HES APC are available through the Data Access Request Service at NHS Digital.[28]

The linkage was conducted using SQL Server Management Studio V.18.4. Additional analyses were conducted using STATA MP V.15.1, with the significance level set at 0.05.

## RESULTS

The overall matching for a unique person within the full NDTMS sample (n=268 251) to a HES APC hospitalisation record generated n=213 814 linked records, representing a linkage rate of 79.7%. The proportion linked according to the matching stages described earlier were:

stage 1: 10.7%, stage 2: 5.7%, stage 3: 72.5% and stage 4: 11.1%.

## Gold-standard subsample

The overall matching for a unique person within NDTMS to a HES APC hospitalisation record using the 'gold-standard' subset of people with an NHS number available in NDTMS, generated n=1153 linked records using NHS number, representing a linkage rate of 86.6%. Using the four-stage algorithm within the 'gold-standard' population generated n=1053 linked records with a linkage rate of 79.3%. Although this was lower than the NHS number match rate this suggests that the majority of unlinked records were true non-links (ie, individuals who had not previously been hospitalised) and not missed matches. Of the n=1053 records linked using the four-stage algorithm, 102 were not linked by the gold standard. These included n=36 records that disagreed on NHS number and were therefore assumed to be false links, and 66 that had missing or invalid NHS numbers and could represent either false links or links missed by the gold-standard. These two possibilities suggest a precision of between 90.3% and 96.4%, respectively. Of the n=1153 records matched using the NHS number n=202 were not matched by the four-stage algorithm suggesting a sensitivity of between 82.5% and 83.3%, respectively.

Table 1 summarises the associations between sociodemographic and clinical variables and linkage by four-stage algorithm within the gold-standard subsample who linked to HES APC via their NHS number (n=1153). Within this sample we compared individuals who were classified as linked or non-linked using the four-stage algorithm, an adjusted OR (aOR) greater than 1 denoting increased odds of successful linkage when compared with the reference value. In the adjusted model, we found significant differences in the odds of linking for sex, age and ethnicity. There was strong evidence that when compared with women, men were significantly less likely to link to HES APC (aOR 0.48, 95% CI 0.30 to 0.79, p=0.003), when compared with those aged between 18 and 30, those aged between 46 and 60 were significantly more likely to link (aOR 2.28, 95% CI 1.08 to 4.82, p=0.03), and when compared with people of a white ethnicity, people with a non-white ethnicity were significantly less likely to link (aOR 0.35, 95% CI 0.20 to 0.63, p<0.001). The multilevel model was significantly superior to the fixed-effects logistic model (p<0.001), with an intraclass correlation coefficient (ICC) of 0.13 (95% CI 0.04 to 0.36).

## Analysis of linkage error

Weighting the probability of being linked to HES APC data using the four-stage algorithm demonstrated a correction of linkage bias within the 'gold-standard' sample, the results of which are summarised in online supplemental table S2.

The full linked sample had a total of 1 624 152 inpatient hospital admissions since HES database inception in April 1997 until January 2020, with a total time spent in

**Table 1** Sociodemographic and clinical characteristics of the n=1153 individuals in NDTMS linked to Hospital Episode Statistics Admitted Patient Care (HES APC) using National Health Service (NHS) number characterised as either linked or non-linked to HES APC using the four-stage algorithm

| | | Linked pairs n (%) | Non-linked residuals n (%) | OR (95% CI) for positive linkage | P value | aOR† (95% CI) for positive linkage | P value |
|---|---|---|---|---|---|---|---|
| **All** | **All** | 951 (82.5) | 202 (17.5) | – | – | – | – |
| Sociodemographic | **Sex** | | | | | | |
| | Female | 320 (86.5) | 50 (13.5) | Reference | – | Reference | – |
| | Male | 631 (80.6) | 152 (19.4) | 0.66 (0.46 to 0.95) | 0.03* | 0.48 (0.30 to 0.79) | 0.003* |
| | **Age in years (at presentation to D&A services)** | | | | | | |
| | 18–30 | 97 (81.5) | 22 (18.5) | Reference | – | Reference | – |
| | 31–45 | 497 (79.1) | 131 (20.9) | 0.79 (0.46 to 1.36) | 0.40 | 1.10 (0.56 to 2.19) | 0.78 |
| | 46–60 | 336 (87.7) | 47 (12.3) | 1.53 (0.85 to 2.76) | 0.16 | 2.28 (1.08 to 4.82) | 0.03* |
| | 60+ | 21 (91.3) | 2 (8.7) | 1.62 (0.34 to 7.83) | 0.55 | 1.79 (0.33 to 9.62) | 0.50 |
| | **Deprivation (IMD) quintile** | | | | | | |
| | First (most deprived) | 328 (81.6) | 74 (18.4) | Reference | – | Reference | – |
| | Second | 275 (85.1) | 48 (14.9) | 0.96 (0.62 to 1.49) | 0.87 | 0.81 (0.50 to 1.32) | 0.40 |
| | Third | 210 (90.5) | 22 (9.5) | 1.38 (0.77 to 2.47) | 0.28 | 1.04 (0.55 to 1.98) | 0.90 |
| | Fourth | 89 (89.9) | 10 (10.1) | 1.07 (0.49 to 2.34) | 0.87 | 1.22 (0.46 to 3.23) | 0.69 |
| | Fifth (least deprived) | 17 (85.0) | 3 (15.0) | 1.14 (0.30 to 4.30) | 0.85 | 0.49 (0.12 to 1.99) | 0.32 |
| | **Residential status‡** | | | | | | |
| | Non-NFA postcode | 913 (81.9) | 202 (18.1) | Reference | | Reference | |
| | NFA postcode | 38 (100.0) | 0 (0.0) | – | – | – | – |
| | **Ethnicity§** | | | | | | |
| | White | 840 (88.9) | 105 (11.1) | Reference | | Reference | |
| | Non-white | 80 (73.4) | 29 (26.6) | 0.42 (0.24 to 0.71) | <0.001* | 0.35 (0.20 to 0.63) | <0.001* |
| Clinical | **Substance misuse¶** | | | | | | |
| | Opioid | 503 (77.3) | 148 (22.7) | Reference | | Reference | |
| | Alcohol only | 311 (90.7) | 32 (9.3) | 2.41 (1.57 to 3.68) | <0.001* | 1.57 (0.95 to 2.61) | 0.08 |
| | Non-opioid and alcohol | 75 (91.5) | 7 (8.5) | 2.16 (0.95 to 4.93) | 0.07 | 2.03 (0.76 to 5.44) | 0.16 |
| | Non-opioid only | 62 (80.5) | 15 (19.5) | 1.12 (0.59 to 2.12) | 0.73 | 0.98 (0.44 to 2.18) | 0.96 |

*p<0.05.
†Adjusted for all other covariates listed in table.
‡Residential status was omitted from the model as all people with an NFA postcode were linked.
§Office of Population Censuses and Surveys (OPCS) categories A, B and C collapsed as white, all other OPCS categories (D–S) collapsed as non-white.
¶NDTMS categorisation *opioid*: clients with any mention of opioid use in any treatment episode during the year irrespective of other substances cited; *alcohol only*: clients who present with problems related to alcohol but no other substances; *non-opioid and alcohol*: clients with non-opioid drug *and* alcohol use problems (but *not* opioids) recorded in any treatment episode during the year; *non-opioid only*: clients who present for treatment related to non-opioid drug use but *not* opioids or alcohol.
aOR, Adjusted OR; D&A, Drug and Alcohol; IMD, Index of Multiple Deprivation; NDTMS, National Drug Treatment Monitoring System; NFA, no fixed abode.

hospital of 14 461 years, and an overall average length of hospital stay of 3 days. Table 2 summarises the associations between sociodemographic and clinical variables and the average length of hospital admission for linked individuals differentiated into those with an average length of hospital admission <5 days and those with an average length of hospital admission ≥5 days. An aOR greater than 1 denotes increased odds of an average length of hospital admission ≥5 days when compared with the reference value. In the adjusted model, we found significant differences in the average length of hospital admission across the majority of studied sociodemographic and clinical factors. There were no substantial differences between the estimates generated from the adjusted models following inverse probability or sufficient matching data weighting. The multilevel model was significantly superior to the fixed-effects logistic model (p<0.001), with an ICC of 0.02 (95% CI 0.01 to 0.02).

## DISCUSSION

Using deterministic matching, a national longitudinal and cross-sectional dataset was built between NDTMS specialist community drug and alcohol treatment data and HES hospitalisation data in England, providing a linkage for 213 814 adults (79.7% of the full NDTMS cohort) to their inpatient hospital records. Using our linkage algorithm there were significant differences in the sociodemographic and clinical characteristics between the linked and non-linked samples, with individuals more likely to link if they were women, white, and aged between 46 and 60 years old. Using the linked data, we were able to demonstrate that individuals were more likely to have an increased average length of hospital stay if they were male, older, had no fixed residential address, and had problematic opioid use. These effects did not change substantially following inverse probability weighting, suggesting they were not driven by bias from linkage error.

### Analysis of linkage biases

Very few studies have examined linkage error in the context of people with substance use disorders. Using our deterministic algorithm, n=54 437 (20.3%) of individuals were not linked to HES APC hospitalisation records. Linkage of the gold-standard sample suggests that approximately two-thirds of these are true non-links (ie, arising because the individual had never been hospitalised and therefore had no HES record), and the remaining third are missed matches. When using the 'gold-standard' sample 86.6% of records matched using NHS number, as such 86.6% is likely to estimate the overall true match rate. We can thus infer that a roughly similar percentage of the n=10 011 NDTMS records with insufficient matching data should match and are therefore genuine missed-matches in the total sample (n=8670). Based on our linkage sensitivity these 8670 records constitute just under half of the total number of likely missed match records when using the four-stage algorithm. The sociodemographic

and clinical characteristics of this cohort can be found in online supplemental table S3, and when compared with the NDTMS cohort with sufficient matching data, demonstrate a substantially lower odds of having sufficient matching data if individuals were men, younger and problematic opioid users. This indicates a higher likelihood of missed matches within these groups which is in accordance with the reduced odds of linkage for men, younger and non-white individuals observed using the four-stage algorithm in the 'gold-standard' sample.

We found that older age groups were more likely to link which may reflect a greater availability of accurate personal identifiers in the records of this population as by living longer they have had greater potential exposure to drug and alcohol services compared with other age groups, and an increased number of hospitalisation records, and therefore potentially more values of matching variables. Previous research has suggested that individuals from black and ethnic minorities are more likely to have administrative records with inaccurately recorded dates of birth and higher levels of residential instability, which may be applicable to this sample, and partially account for the reduced likelihood in of linkage compared with white individuals.[29] It is reassuring however that in our sample, linkage biases do not appear to have significant effect on the associations between substance misuse and average length of hospital stay.

### Strengths and limitations of the matching methods and evaluation

This represents the first study of its kind to link centralised national level substance misuse treatment data and inpatient hospitalisation records, and provides an example of how potential non-random loss between routinely collected administrative datasets can be adjusted for by weighting techniques.[30] As we had access to complete source data records, we were able to demonstrate that linkage error did not appear to lead to systematic bias and misestimation of sociodemographic and clinical factor associations with average length of hospital stay. It should be noted that in order to evaluate potential linkage bias within this paper we only report a single healthcare outcome. Following evaluation for potential linkage bias interrogation of the resultant dataset will be possible to address a number of key research and policy questions. There are also a number of limitations. Due to the previous practice of the UK Home Office compiling full names and addresses of all registered addicts in its 'Index of Addicts',[31] and more generally the stigma experienced by people with substance use disorders, NDTMS is careful to collect only the minimum amount of personal identifier information it deems necessary to balance the need for population surveillance, with legitimate concerns about individual identification. An unfortunate consequence, however, is that no single unique identifier, such as NHS number, is routinely collected within NDTMS and the personal identifiers which are collected are typically fewer than in other UK government held datasets.

**Table 2** The odds of an average ≥5 days length of hospital admission in the 213 814 people in treatment at drug and alcohol services

| | | ≥5 days average hospital admission length n (%) | <5 days average hospital admission length n (%) | OR (95% CI) | aOR* (95% CI) | Inverse probability weighted for missed links aOR† (95% CI) | Weighted for sufficient matching data aOR (95% CI) |
|---|---|---|---|---|---|---|---|
| All | **All** | 25 814 (12.1) | 188 000 (87.9) | | | | |
| Sociodemographic | **Sex** | | | | | | |
| | Female | 6715 (9.4) | 65 114 (90.6) | Reference | Reference | Reference | Reference |
| | Male | 19 099 (13.5) | 122 886 (86.5) | 1.51 (1.47 to 1.56) | 1.42 (1.37 to 1.46) | 1.42 (1.37 to 1.47) | 1.42 (1.37 to 1.47) |
| | **Age in years (at presentation to D&A services)** | | | | | | |
| | 18–30 | 1979 (7.1) | 26 040 (92.9) | Reference | Reference | Reference | Reference |
| | 31–45 | 10 804 (10.9) | 88 392 (89.1) | 1.58 (1.50 to 1.66) | 1.46 (1.38 to 1.54) | 1.46 (1.37 to 1.54) | 1.46 (1.37 to 1.54) |
| | 46–60 | 10 771 (14.7) | 62 634 (85.3) | 2.17 (2.06 to 2.28) | 2.08 (1.96 to 2.20) | 2.07 (1.95 to 2.20) | 2.08 (1.96 to 2.20) |
| | 60+ | 2260 (17.1) | 10 934 (82.9) | 2.62 (2.46 to 2.80) | 2.84 (2.65 to 3.06) | 2.81 (2.61 to 3.04) | 2.84 (2.63 to 3.06) |
| | **Deprivation (IMD) quintile** | | | | | | |
| | First (most deprived) | 8340 (12.4) | 58 831 (87.6) | Reference | Reference | Reference | Reference |
| | Second | 6988 (12.4) | 49 543 (87.6) | 0.99 (0.95 to 1.03) | 1.01 (0.97 to 1.05) | 1.01 (0.97 to 1.06) | 1.01 (0.97 to 1.06) |
| | Third | 4560 (11.5) | 35 324 (88.5) | 0.94 (0.90 to 0.98) | 0.98 (0.94 to 1.02) | 0.98 (0.93 to 1.04) | 0.98 (0.93 to 1.04) |
| | Fourth | 3045 (10.8) | 25 250 (89.2) | 0.90 (0.86 to 0.95) | 0.95 (0.90 to 1.00) | 0.95 (0.90 to 1.01) | 0.95 (0.90 to 1.01) |
| | Fifth (least deprived) | 1289 (10.0) | 11 599 (90.0) | 0.85 (0.79 to 0.91) | 0.90 (0.84 to 0.97) | 0.90 (0.84 to 0.97) | 0.90 (0.85 to 0.97) |
| | **Residential Status** | | | | | | |
| | Non-NFA postcode | 24 108 (11.8) | 180 389 (88.2) | Reference | Reference | Reference | Reference |
| | NFA postcode | 1706 (18.3) | 7611 (81.7) | 1.65 (1.57 to 1.75) | 1.44 (1.27 to 1.63) | 1.43 (1.24 to 1.65) | 1.43 (1.24 to 1.65) |
| | **Ethnicity‡** | | | | | | |
| | White | 22 808 (11.8) | 170 772 (88.2) | Reference | Reference | Reference | Reference |
| | Non-white | 2462 (15.3) | 13 650 (84.7) | 1.20 (1.14 to 1.26) | 1.19 (1.13 to 1.25) | 1.18 (1.11 to 1.26) | 1.19 (1.12 to 1.27) |
| | **Substance misuse§** | | | | | | |
| Clinical | Opioid | 15 309 (14.3) | 92 102 (85.7) | Reference | Reference | Reference | Reference |
| | Alcohol only | 6776 (10.4) | 58 084 (89.6) | 0.71 (0.69 to 0.74) | 0.68 (0.66 to 0.70) | 0.68 (0.65 to 0.72) | 0.68 (0.64 to 0.71) |
| | Non-opioid and alcohol | 2055 (9.0) | 20 820 (91.0) | 0.58 (0.55 to 0.61) | 0.67 (0.63 to 0.70) | 0.67 (0.63 to 0.72) | 0.67 (0.62 to 0.71) |
| | Non-opioid only | 1674 (9.0) | 16 994 (91.0) | 0.59 (0.56 to 0.62) | 0.74 (0.70 to 0.79) | 0.76 (0.70 to 0.82) | 0.74 (0.69 to 0.80) |

*Adjusted for all other covariates listed in table.
†Adjusted model with inverse probability weighting for matching included.
‡Office of Population Censuses and Surveys (OPCS) categories A, B and C collapsed as white, all other OPCS categories (D–S) collapsed as non-white.
§NDTMS categorisation *opioid*: clients with any mention of opioid use in any treatment episode during the year irrespective of other substances cited; *alcohol only*: clients who present with problems related to alcohol but no other substances; *non-opioid and alcohol*: clients with non-opioid drug *and* alcohol use problems (but *not* opioids) recorded in any treatment episode during the year; *non-opioid only*: clients who present for treatment related to non-opioid drug use but *not* opioids or alcohol.
aOR, Adjusted OR; D&A, Drug and Alcohol; IMD, Index of Multiple Deprivation; NFA, no fixed abode.

This creates a unique problem for NDTMS data linkage, which is compounded by the fact that, when compared with the general population, individuals within NDTMS are also less likely to be registered with a GP, more likely to not have a residential address, and potentially have an interest in providing non-accurate personal identification information to drug and alcohol services. All of the above reasons may contribute to the observed increased rate of false and missed matches compared with other national data linkages.[30] Nevertheless, a national centralised data repository for substance misuse treatment presents a unique opportunity to link with other health and social care record systems, provided consent is given by those individuals, in an attempt to improve the lives of people with substance use disorders. This paucity of available personal identifiers results in an increased risk of both false and missed matches, particularly at lower confidence matching stages and these limitations could have led to our match rate being an overestimation of the linkage performance. As such in order to minimise the risk of false matches, records that linked with multiple different unique records from either dataset were removed and treated as non-links. These could reflect imperfect internal linkage or deduplication of NDTMS or HES; that is, these could be true multiple links and this linkage strategy could increase the rate of missed matches. When 'gold-standard' data are used to assess linkage quality this is assumed to be representative in terms of the distribution of the quality of matching and analysis variables. Although our 'gold-standard' dataset unique person identifier is NHS number we cannot exclude the possibility there may be coding errors within NHS numbers and the dataset may not represent the remainder of records. Although there was a significantly lower linkage rate using our algorithm compared to using NHS number within the gold-standard sample (79.7% vs 86.6%), differences in ethnicity and age between the full and gold-standard NDTMS samples partially explain this difference, but do not appear to contribute significant bias due to linkage error.

## Implications

This linkage between substance misuse treatment and hospitalisation records offers a new powerful tool to evaluate the impact of specialist treatment on alcohol and drug related harm in England. Through its interrogation, and via additional sanctioned linkage to datasets from other government departments, for example, the Department for Work and Pensions, these data may hopefully be able to provide insight and knowledge to improve the lives of people with substance use disorders. While biases due to linkage error may produce misleading results in our sample, linkage biases appear to have little effect on the association between drug and alcohol treatment and length of hospital admission. However, without ongoing ability to probe information within the source data, potential linkage error could be introduced without future analysts being aware that there was need for it to be accounted for.

In time, we hope this resource will generate a wide network of granular data and analytical expertise, which can be used to inform both commissioning and service provision to better meet the needs of people with substance use disorders in England. The immediate next steps are to evaluate the most common reasons for hospital admission within the cohort of people accessing drug and alcohol treatment and to assess the impact of engagement in, and successful completion of, drug and alcohol treatment on individual and national rates hospitalisation. It is important to note that as subsequent analyses of the resultant linked dataset are conducted, any potential bias associated with the linkage process should always be considered in the interpretation of any findings.

**Author affiliations**
[1]National Addiction Centre, Institute of Psychiatry, Psychology and Neuroscience King's College London, London, United Kingdom
[2]Deaprtment of Psychological Medicine, Institute of Psychiatry, Psychology and Neuroscience King's College London, London, United Kingdom
[3]South London and the Maudsley NHS Foundation Trust, London, United Kingdom
[4]Public Health England, London, United Kingdom
[5]Intensive Care National Audit & Research Centre, London, United Kingdom
[6]Great Ormond Street Institute of Child Health, University College London, London, United Kingdom

**Contributors** All authors meet the ICMJE criteria for authorship. ER: formulated the research question, designed and carried out the study, analysed the data and drafted the article. JD: contributed to the study design, data analysis and writing the article. KH: contributed to the study design, data analysis and writing the article. MH: contributed to the formulation of the research question, study design, data analysis and writing the article. JK: contributed to the study design, data interpretation and writing the article. MW: contributed to the study design, data interpretation and writing the article. BE: contributed to the formulation of the research question, study design, data analysis and writing the article. CD: contributed to the formulation of the research question, study design, data analysis and writing the article.

**Funding** This paper represents independent research funded by the Medical Research Council (MRC), as part of the corresponding author's MRC Addiction Research Clinical (MARC) Fellowship. The research was part funded by the NIHR Biomedical Research Centre at South London and Maudsley NHS Foundation Trust and King's College London, and by the NIHR Collaboration for Leadership in Applied Health Research and Care South London (NIHR CLAHRC South London) now recommissioned as NIHR Applied Research Collaboration South London, and both CD and MH receive funding from an NIHR Senior Investigator award. KH is funded in part by the Wellcome Trust (grant number 212953/Z/18/Z) and benefits from infrastructure funded by NIHR GOSH BRC and Health Data Research UK.

**Disclaimer** The funders had no contribution to the study design; in the collection, analysis, and interpretation of data; in the writing of the report; and in the decision to submit the article for publication. All authors were independent from funders had full access to all of the data (including statistical reports and tables) in the study and take responsibility for the integrity of the data and the accuracy of the data analysis. The views expressed are those of the authors and not necessarily those of the MRC, the National Health Service (NHS), the NIHR, Public Health England (PHE) or the Department of Health and Social Care (DHSC).

**Competing interests** MH is principal investigator of RADAR-CNS consortium—a public private partnership in collaboration with five pharma companies—Janssen, Biogen, UCB, MSD and Lundbeck, outside of the submitted work.

**Patient consent for publication** Not required.

**Ethics approval** Approval to conduct the linkage was granted under regulation 3 of the Health Service (Control of Patient Information), Regulations 2002 following review by the PHE Caldicott Advisory Panel (CAP) (Ref: CAP-2019-06).

**Provenance and peer review** Not commissioned; externally peer reviewed.

**ORCID iDs**
Emmert Roberts http://orcid.org/0000-0002-4152-5570
Katie L Harron http://orcid.org/0000-0002-3418-2856

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
