## [Reviewer comments · BMJ Open]

ARTICLE DETAILS

TITLE (PROVISIONAL)	A national administrative record linkage between specialist community drug and alcohol treatment data (The National Drug Treatment Monitoring System (NDTMS)) and inpatient hospitalisation data (Hospital Episode Statistics (HES)) in England; Design, Method and Evaluation
AUTHORS	Roberts , Emmert; Doidge, James; Harron, Katie; Hotopf, Matthew; Knight, Jonathan; White, Martin; Eastwood, Brian; Drummond, Colin

VERSION 1 – REVIEW

REVIEWER	Sally Nathan UNSW Sydney, Australia
REVIEW RETURNED	29-Aug-2020

GENERAL COMMENTS	I am a public health social scientist leading a team undertaking a drug and alcohol treatment linkage study in Australia, but am not a statistician. I found the article informative for those undertaking linkage studies to consider issues of precision and sensitivity in these analyses. This manuscript reports important research in the field of drug and alcohol treatment evaluation about how to examine the relationship between treatment and other health care utilisation, such as hospitalisations. It also identifies and discusses possible limitations of the approach at the matching stage and statistical methods to address them. The one addition that would improve the article would be to include further detail about patient and public involvement, either in the manuscript or as supplementary material. Data linkage, with or without consent, needs to have public confidence in the security of data and the value of such research as a public good to inform better service delivery for people with drug and alcohol issues. I wanted to know more about how the Carer Advisory Group add Treatment Expert Group were informed about the study and their contributions in the current analysis presented and/or future work planned with the linked data set.
---

REVIEWER	Maximilian Gahr University Hospital of Ulm, Department of Psychiatry and Psychotherapy
REVIEW RETURNED	19-Oct-2020

GENERAL COMMENTS	This is a very interesting and important paper. Substance misuse treatment and hospitalisation records were successfully linked. Now, this approach has to be tested in other health care systems.
--

VERSION 1 – AUTHOR RESPONSE

Response to Reviewer Comments

Reviewer: 1

Reviewer Name: Sally Nathan

Institution and Country: UNSW Sydney, Australia

Please state any competing interests or state 'None declared': None declared

I am a public health social scientist leading a team undertaking a drug and alcohol treatment linkage study in Australia, but am not a statistician. I found the article informative for those undertaking linkage studies to consider issues of precision and sensitivity in these analyses. This manuscript reports important research in the field of drug and alcohol treatment evaluation about how to examine the relationship between treatment and other health care utilisation, such as hospitalisations. It also identifies and discusses possible limitations of the approach at the matching stage and statistical methods to address them.

Thank you for your comments

The one addition that would improve the article would be to include further detail about patient and public involvement, either in the manuscript or as supplementary material. Data linkage, with or without consent, needs to have public confidence in the security of data and the value of such research as a public good to inform better service delivery for people with drug and alcohol issues. I wanted to know more about how the Carer Advisory Group and Treatment Expert Group were informed about the study and their contributions in the current analysis presented and/or future work planned with the linked data set.

Thank you for your comments. We have amended and expanded the Patient and Public Involvement section as suggested on Page 7 to state "The study benefited throughout from discussion with the South London and the Maudsley (SLaM) Biomedical Research Centre (BRC) Data Linkage Service User and Carer Advisory Group, and the PHE Alcohol Treatment Expert Group which includes experts with lived experience. The former group represents a regular meeting of people whom have an interest in projects involving data linkage, and who have lived experience of mental health diagnoses, including substance use disorders. They receive on-going training on data matching processes, and hence can make recommendations on the acceptability of suggested data flows. The current proposal was presented in June 2018, and there was group-wide acknowledgement of the importance of the proposed linkage, based on personal experience of treatment experiences in drug and alcohol services. The group were content with the linkage methodology proposed, including the use of patient identifiers. Both groups will remain involved in subsequent analysis plans from any resultant linked data."

Reviewer: 2

Reviewer Name: Maximilian Gahr

Institution and Country: University Hospital of Ulm, Department of Psychiatry and Psychotherapy, Germany

Please state any competing interests or state 'None declared': None.

This is a very interesting and important paper. Substance misuse treatment and hospitalisation records were successfully linked. Now, this approach has to be tested in other health care systems.

Thank you for your comments.

VERSION 2 – REVIEW

REVIEWER	Sally Nathan UNSW Sydney
REVIEW RETURNED	03-Nov-2020

GENERAL COMMENTS	Thank you for adding the details about the role of consumers and experts. It is really important that we share processes for engaging with service users in data linkage studies. The only suggestion is to find another word other than 'content' in last sentence of the additional text - maybe satisfied or supportive? Best wishes
---